# Inner Shell of the Chestnut (*Castanea crenatta*) Suppresses Inflammatory Responses in Ovalbumin-Induced Allergic Asthma Mouse Model

**DOI:** 10.3390/nu14102067

**Published:** 2022-05-15

**Authors:** Chang-Yeop Kim, Jeong-Won Kim, Jin-Hwa Kim, Ji-Soo Jeong, Je-Oh Lim, Je-Won Ko, Tae-Won Kim

**Affiliations:** 1BK21 FOUR Program, College of Veterinary Medicine, Chungnam National University, 99 Daehak-ro, Daejeon 34131, Korea; 963ckdduq@gmail.com (C.-Y.K.); lilflflb@gmail.com (J.-W.K.); jinhwa926@g.cnu.ac.kr (J.-H.K.); jisooj9543@gmail.com (J.-S.J.); 2BK21 FOUR Program, College of Veterinary Medicine, Chonnam National University, 77 Yongbong-ro, Buk-gu, Gwangju 61186, Korea; dvmljo@gmail.com

**Keywords:** asthma, chestnut inner shell extract, airway inflammation, inducible nitric oxide synthase, cyclooxygenase-2, matrix metalloproteinase-9

## Abstract

The inner shell of the chestnut (Castanea crenata) contains various polyphenols, which exert beneficial biological effects. Hence, we assessed the anti-inflammatory efficacy of a chestnut inner shell extract (CIE) in ovalbumin (OVA)-induced allergic asthma. We intraperitoneally injected 20 μg of OVA with 2 mg of aluminum hydroxide on days 0 and 14. On test days 21, 22, and 23, the mice were treated with aerosolized 1% (*w/v*) OVA in saline. CIE was administered orally at 100 and 300 mg/kg on days 18–23. CIE significantly reduced inflammatory cytokines and cells and immunoglobulin-E increased by OVA. Anti-inflammatory efficacy was revealed by reduction of inflammatory cell migration and mucus secretion in lung tissue. Further, CIE suppressed the OVA-induced nuclear factor kappa B (NF-κB) phosphorylation. Accordingly, the expression of cyclooxygenase (COX-2), inducible nitric oxide synthase (iNOS), and matrix metalloproteinase-9 (MMP-9) were decreased sequentially in lung tissues. CIE alleviated OVA-induced airway inflammation by restraining phosphorylation of NF-κB and the sequentially reduced expression of iNOS, COX-2, leading to reduced MMP-9 expression. These results indicate that CIE has potential as a candidate for alleviating asthma.

## 1. Introduction

At present, an estimated 300–400 million people have asthma, which is an increasingly important pulmonary disease characterized by mucus hypersecretion, inflammation, and airway remodeling [1,2]. The risk of developing asthma has increased due to increased air pollution, stress and lack of nutrient synthesis due to increased internal activity [3]. Because asthma is a chronic inflammatory illness, systemic corticosteroids are widely used both intermittently and long-term as a treatment for this but can cause numerous side effects [4]. For this reason, it is necessary to search candidate substances that can supplement this treatment. The important causes of these pathophysiological changes are increased type 2 cytokines interleukin (IL)-4, -5, and -13 [5]. These mediators are implied in inflammatory cell migration and immunoglobulin (Ig)E elevation, resulting in mucus hypersecretion and airway remodeling [6]. Furthermore, signaling molecules related to elevated type 2 cytokines, causing the progression of asthma, are well established from many previous studies. They include inducible nitric oxide synthase (iNOS), cyclooxygenase-2 (COX-2), and nuclear factor kappa B (NF-κB) [7,8]. Another important mediator involved in asthma is the family of extracellular proteases represented by matrix metalloproteinase 9 (MMP-9), causing tissue remodeling through destruction of extracellular structures [9]. Therefore, suppression of these factors is an important step in controlling asthma.

The chestnut inner shell (CIE) has long been used topically as a cosmetic in East Asia [10]. CIE exhibits curative effects, including antioxidant, anticancer, anti-inflammatory, and anti-wrinkle/firming effects [10]. According to a previous study, CIE has the highest clearance efficiency for reactive oxygen species and enhances the efficacy of endogenous antioxidant enzymes activities [11,12]. In particular, Cerulli et al. [13] reported that CIE has anti-inflammatory effects through inhibiting NF-kB phosphorylation and subsequent inflammatory cytokine production. CIE has an abundance of polyphenolic compounds divided into three sub-components; flavonoids, phenolic acids, and tannins [10]. Gallic acid (a phenolic acid) and ellagic acid (a tannin) are the main components of CIE [14]. Sorice et al. [15] demonstrated that CIE had anticancer effects in HepG2 and contained high levels of gallic acid, followed by ellagic and syringic acids. Based on previous studies and the HPLC results from this study, we hypothesize that CIE can offer protection against airway inflammatory responses.

This study explored the anti-asthmatic effect of CIE in asthmatic mice by measuring the degree of decrease in inflammatory cell migration, cytokine levels, inflammatory response and mucus secretion increased by OVA. Specifically, we evaluated the anti-inflammatory mechanisms associated with CIE by focusing on changes in NF-κB, COX-2, iNOS, and MMP-9.

## 2. Materials and Methods

### 2.1. Preparation of Chestnut Inner Shell Extract

A standardized chestnut inner shell extract was prepared according to our previous study [16,17]. In brief, the inner shell of Castanea crenata (Shingongju chestnut agricultural cooperative, Gong-ju, Korea) was dried at 40 °C and ground into powder. In total, 300 g of powder was incubated in 3 L of 70% ethanol. After 24 h, the aqueous layer was collected and freeze-dried. The freeze-dried extract was subjected to a prepared open column to exclude the water-soluble fraction. The extract was loaded onto a Diaion HP-20 (Mitsubishi Chemical, Tokyo, Japan) filled open column, and the water washing step was followed by ethanol elution. The elute was freeze-dried and subjected to HPLC analysis for standardization.

### 2.2. HPLC Analysis

The HPLC analysis was performed as previously described with minor modifications [16,17]. HPLC analysis used an HPLC 1260 Infinity II (Agilent Technologies, Inc., Santa Clara, CA, USA) system with an Eclipse C18 column (2.1 × 150 mm, 3.5 μm; Agilent Technologies) under 40 °C. Ellagic acid (>95% purity, Sigma-Aldrich, CA, USA) was used as a marker. The mobile phases were 0.1% formic acid (A) in DW and acetonitrile (B). A flow rate of 0.12 mL/min was used, with the gradient condition as flow with a of flow rate: 95–70% A, 0–30 min; 70–5% A, 30–56 min; 30–0% B, 56–57 min; 0–95% B, 57–58 min; and 95% A, 58–70 min. The injection volume was 2 μL, and the UV wavelength was set at 275 nm. Data acquisition was performed by using the Agilent OpenLab CDS ChemStation (version 2.15.26; Santa Clara, CA, USA).

### 2.3. Animals

Female BALB/c mice (6 weeks old) were acquired from Orient Co. (Seoul, Korea). All mice were housed under standard laboratory conditions (55 ± 5% humidity, 23 ± 2 °C, and 12 h of light/dark cycle) with ad libitum access to sterilized water and standard lab chow (Orientbio, Seongnam, Korea). Animal procedures were approved by the Chungnam National University Animal Care and Use Committee and performed in accordance with established animal care, use, and experimental guidelines.

### 2.4. Experimental Procedure

Mice were allocated into groups of six mice each. The groups were NC (normal control; PBS treatment), OVA (OVA treatment), DEX (OVA treatment + DEX 3 mg/kg, p.o.), and CIE L or H (OVA treatment + CIE 100 or 300 mg/kg, p.o., respectively). The allergic asthma mouse model was induced through OVA using the method described in our previous study [18]. Briefly, each mouse was immunized by 20 μg OVA (emulsified in 10 mg/mL of aluminum hydroxide and PBS) intraperitoneal injection on test days 0 and 14. On test days 21, 22, and 23, the mice were treated with aerosolized 1% (*w/v*) OVA in saline, which was administered 3 times a day for 20 min by an Omron nebulizer (NE-U12; Omron, Tokyo, Japan). Dexamethasone (10 mg/kg) and CIE (100 mg/kg and 300 mg/kg) were orally administered from days 18–23. During oral administration, 1% OVA (*w/v*) solution was made into an aerosol with an ultrasonic nebulizer and exposed to mice to stimulate the immune system three times a day for 20 min on test days 21, 22, and 23. All animals were necropsied 48 h after the last OVA nebulizing treatment (test day 25).

### 2.5. Airway Hyperresponsiveness (AHR)

AHR was assessed 24 h after the last OVA challenge by whole-body volumetric (OCP3000, Allmedicus, Seoul, Republic of Korea). Each animal was placed in a standard plastic room and exposed to aerosolized methylcholine (0, 10, 20, and 30 mg/mL; Sigma-Aldrich, St. Louis, MO, USA) via a nebulizer for 3 min. The enhanced pause (Penh) value was measured after each methylcholine exposure.

### 2.6. Analysis of Immunoglobulin E (IgE) OVA-Specific IgE in Serum

The animals were sacrificed 24 h after calculating AHR, and blood was gathered from the postcaval vein. The serum-containing supernatant was obtained after centrifugation at 800× *g* for 20 min. The serum levels of IgE were evaluated using an enzyme-linked immunosorbent assay (ELISA) kit (BioLegend, San Diego, CA, USA) according to the supplier’s instructions. Absorbance was detected at 450 nm using a plate reader of ELISA (Bio-Rad Laboratories, Hercules, CA, USA).

### 2.7. Analysis of Bronchoalveolar Lavage Fluid (BALF)

Following blood collection, BALF was collected from the mice and processed as previously described [19]. Briefly, after the tracheotomy, an endotracheal syringe was intubated into the trachea. After ice-cold PBS (0.7 mL) was injected into the lungs and recovering it twice (1.4 mL of total volume), BALF was centrifuged at 800× *g* for 10 min, and the supernatant was stored at –80 °C for analysis of type 2 cytokines. To distinguish cell types, the pellet was resuspended and the suspended cells were attached to the slide using cytospin (Hanil Science Industrial, Seoul, Republic of Korea) (200× *g* at 4 °C for 10 min). Cells on dried slides were fixed and stained with Diff-Quik^®^ reagent (Sysmex Co., Kobe, Japan). The levels of the type 2 cytokines interleukin-4, -5, and -13 were determined via a commercial enzyme-linked immunosorbent assay (ELISA, RayBi-othech Life, Incorporation, Georgia Norcross) and detected at 450 nm using a plate reader of ELISA (Bio-Rad Laboratories).

### 2.8. Histopathological Analysis of the Lung Tissue

First, after obtaining a BALF sample, the left lung tissue was isolated and stored in 10% (*v/v*) neutral buffered formalin. After going through the general slide production process (tissue dehydration, paraffin embedding), hematoxylin and eosin (H&E; TissuePro Technology, Gainesville, FL, USA) and periodic acid-Schiff (PAS; IMEB Inc., San Marcos, CA, USA) staining was performed after tissue was attached to the slide with a thickness of 4 μm. An image analyzer was used to quantify airway inflammation and mucus production (IMT i-Solution software, Vancouver, BC, Canada).

### 2.9. Western Blot Analysis

Protease and phosphatase inhibitors (Roche, Basel, Switzerland) were added to tissue lysis/extraction reagent (Sigma-Aldrich, St. Louis, MO, USA), and the lung tissue was homogenized with this solution (1/10 *w/v*). Protein concentrations of each sample were detected using BCA (Thermo Fisher Scientific, Waltham, MA, USA). Western blotting was performed as described previously [20], and the following antibodies were used: iNOS (1:000; Cell Signaling, Danvers, MA), COX-2 (1:000; Cell Signaling), NF-kB (1:000; 1:1000; Abcam, Cambridge, UK), phosphor-NF-kB (1:1000; Abcam, Cambridge, MA, USA) and MMP-9 (1:1000; Abcam, Cambridge, MA, USA). Horse-radish peroxidase-conjugated secondary antibodies (1:10,000; Thermo Fisher Scientific, Waltham, MA, USA) were used and then detected with EZ-Western Lumi Pico (Dogenbio, Seoul, Republic of Korea). The densitometric values of each band were numerically expressed using Chemi-Doc (Bio Rad Laboratories, Hercules, CA, USA).

### 2.10. Analysis of MMP-9 Level in Lung Tissue

As described in the previous paper, MMP-9 activity was measured by the gelatin zymography test method [18]. In principle, blue-stained gel is visualized as a white band by dissolving gelatin where MMP-9 activity is high. The activity of visualized MMP-9 with white band was quantified using Chemi-Doc (Bio Rad Laboratories, Hercules, CA, USA).

The MMP-9 expression in lung tissue was determined using a commercial Immuno-histochemical (IHC) kit (Vector Laboratories, Burlingame, CA) according to the supplier’s protocol. The expression levels of MMP 9 were visualized using 3,3-diaminobenzidine, and the activity of the expressed MMP-9 was quantified using an image analyzer (IMT i-Solution software, Vancouver, BC, Canada).

### 2.11. Statistical Analysis

Data are expressed as the means ± standard deviation (SD). To analyze statistical significance, the data were subjected to analysis of variance followed by multiple comparison tests with Dunnett’s adjustment. Statistical analyses were performed using GraphPad InStat v 3.0 (GraphPad, La Jolla, CA, USA). *p*-values < 0.05 were considered statistically significant.

## 3. Results

### 3.1. Standardization of CIE

The chromatographic separation of CIE was achieved based on our previous study [16,17]. The marker compound was confirmed by comparing the retention time and consistency of the reference peak using UV detection. The retention time for ellagic acid was about 3.1 min, which was also found in CIE chromatography fingerprinting, and the ellagic acid content in CIE was about 0.5 mg/g (Figure 1).

### 3.2. Effects of CIE on AHR in Asthmatic Mice

Compared with the NC group, the OVA group showed significantly increased AHR when exposed to the same concentration of methylcholine (Figure 2). When the CIE group was exposed to a low dose of methylcholine (10 mg/mL), the measured AHR was lower than that in the OVA group with the same concentration, but there was no statistical significance. When measured and exposed to medium (20 mg/mL) and high (30 mg/mL) doses of methylcholine, the CIE group showed a statistically significant decrease in AHR levels compared to the OVA group exposed to the same dose in a dose-dependent manner. The AHR decrease in the CIE groups was similarly observed in the DEX group.

### 3.3. Effects of CIE on Inflammatory Cell Counts in BALF from Asthmatic Mice

In BALF, the number of inflammatory cells found in the OVA group showed a statistically significant increase when compared with the NC group (Figure 3). In the CIE group, a significant decrease in the number of inflammatory cells increased by OVA was observed, and this decrease was particularly pronounced in eosinophils. The results in the CIE groups were similarly observed in the DEX group.

### 3.4. Effects of CIE on the Production of Type 2 Cytokines in BALF and Levels of IgE in Serum from OVA-Induced Asthmatic Mice

IL-4 expression measured in BALF was statistically significantly higher in the OVA group (*p* < 0.01, 58.7 ± 10.44 pg/mL) than in the NC group (Figure 4A), whereas IL-4 levels in BALF measured in the CIE groups (*p* < 0.01, 34.9 ± 6.58 pg/mL in the CIE L group; *p* < 0.01, 35.0 ± 2.88 pg/mL in the CIE H group) were markedly lower than those in the OVA group. Similarly, in mice with OVA-induced asthma, significantly increased IL-5 (*p* < 0.01, 67.5 ± 20.40 pg/mL) and IL-13 (*p* < 0.01, 99.8 ± 22.99 pg/mL) levels were observed in BALF (Figure 4B,C). In all CIE concentration groups, IL-5 (56.7 ± 10.73 pg/mL in the CIE L group; *p* < 0.05, 46.0 ± 8.94 pg/mL in the CIE H group) and IL-13 (*p* < 0.05, 76.9 ± 8.28 pg/mL in the CIE L group; *p* < 0.01, 55.4 ± 4.84 pg/mL in the CIE H group) levels were significantly lower than in the OVA group. Although IL-5 levels decreased in the CIE L group, the effect was not significant.

In the OVA group, total (*p* < 0.01, 139.8 ± 44.1 pg/mL) and OVA-specific (*p* < 0.01, 68.4 ± 16.34 pg/mL) IgE levels were markedly higher than in the NC group. Whereas, the elevated levels of total (*p* < 0.01, 82.3 ± 17.79 pg/mL in the CIE L group; *p* < 0.01, 64.6 ± 13.99 pg/mL in the CIE H group) and OVA-specific (*p* < 0.01, 40.9 ± 9.34 pg/mL in the CIE L group; *p* < 0.01, 31.3 ± 5.29 pg/mL in the CIE H group) IgE were significantly inhibited in the CIE groups, which showed a dose–response relationship (Figure 5). These reductions of type 2 cytokine levels and IgE levels in the CIE groups were similarly observed in the DEX group.

### 3.5. Effects of CIE on Airway Inflammation and Mucus Production in Lung Tissue from OVA-Induced Asthmatic Mice

Significant airway inflammation in the peri-bronchial and perivascular regions in the lung tissues of the OVA group was observed by staining with H&E (Figure 6A,B). However, in the DEX and CIE groups, the migration of inflammatory cells to the peri-bronchial and perivascular areas induced by OVA was significantly reduced.

Similarly, the OVA group showed mucus hypersecretion and hyperplasia of goblet cells in bronchiolar lesions revealed by PAS staining (Figure 6A,C). In contrast, mucus over-production in bronchiolar lesions was significantly less in the DEX and CIE groups than in the OVA group.

### 3.6. Effects of CIE on the Expression of iNOS, COX-2, NF-kB, and MMP-9 in Lung Tissue from OVA-Induced Asthmatic Mice

It was observed that the expression levels of inflammatory proteins COX-2, iNOS, and NF-κB were significantly increased by OVA exposure (Figure 7). The CIE groups exhibited markedly decreased expression levels of COX-2, iNOS, and NF-κB in lung tissue compared with the OVA group, as revealed by Western blots. Further, the expression level of MMP-9 increased by OVA exposure was statistically significantly decreased in the DEX and CIE groups.

### 3.7. Effects of CIE on the Activity and Expression of MMP-9 in Lung Tissue from OVA-Induced Asthmatic Mice

As can be seen from the IHC results (Figure 8A), OVA exposure significantly increased the expression level of MMP-9 in tissues of lung, whereas the increase was lower in the CIE groups showing a dose–response relationship. MMP-9 activity showed results similar to the expression pattern for MMP-9 protein in lung tissue (Figure 8B,C). The zymography results showed that MMP-9 activity in the CIE groups was lower than that in the OVA group showing a dose–response relationship.

## 4. Discussion

Allergic asthma is regarded as one of the important worldwide health issues and has become increasingly important [21]. Eosinophilic airway inflammation, hyperresponsiveness, and mucus hypersecretion are symptoms in the onset of asthmatic asthma [22]. The purpose of the present study was to elucidate, using an OVA-induced asthmatic mice model, the anti-asthmatic efficacy of CIE, which has an abundance of polyphenolic compounds, especially ellagic acid. Although it could be hasty to specify single active compound from chestnut inner shell due to the diversity of contents, the ellagic acid might be a factor responsible for the present beneficial biological effects of CIE. Along with its well-known antiallergic activity, ellagic acid was found to have a relatively higher intestinal absorption rate, which was over 75%, with moderate volume of distribution. Moreover, ellagic acid was reported to satisfy Lipinski’s rule of five, supporting its high drug-likeness [14,23,24]. The present CIE treatment effectively decreased AHR with a reduction in the migration of inflammatory cells and type 2 pro-inflammatory cytokine levels in BALF and IgE levels in serum, which had been elevated by OVA sensitization. These changes were consistent with the results of decreased inflammatory cell migration and decreased hyper mucinogenesis in the lung tissues of CIE-treated mice. Additionally, CIE notably reduced the expression of COX-2 and iNOS and NF-κB phosphorylation followed by reduced MMP-9 expression levels elevated by OVA exposure.

The symptoms of allergic asthma are AHR and increases in inflammatory cells and mucus production [25]. Type 2 cytokines play critical roles in the development of asthma [22]. Cytokines move eosinophils towards the lung lesion to trigger maturation and activation of eosinophils [26]. These events contribute to the production of other inflammatory cytokines, chemokines, and growth factors, resulting in representative symptoms of asthma [27]. Various animal studies have elucidated the role of type 2 cytokines in the pathogenesis of asthma, and this has been demonstrated in several clinical trials [28]. In this study, compared to the OVA group, the CIE groups showed a notable decrease in airway inflammation and mucus production in lung tissue, which is consistent with the AHR results. These reductions were accompanied by an increase in inflammatory cell migration, type 2 cytokine production in BALF, and IgE production in the serum. The CIE groups showed effectively suppressed inflammatory cell migration, especially of eosinophils, in BALF compared with that of the OVA group showing a dose–response relationship. The levels of the pro-inflammatory cytokines IL-4, -5 and 13 expressed in the BALF of the CIE groups showed significantly reduced levels compared to the level expressed in the OVA group and were accompanied by reduced total and OVA-specific IgE levels in the serum. These results suggested that CIE exhibits protective effects against asthmatic responses by reducing type 2 cytokine levels.

NF-κB is a well-known transcription factor frequently involved in the overall inflammatory process, modulating various signaling cascades [29]. Additionally, the role of NF-κB in the pathogenesis of asthma has been well documented in many clinical trials as well as in various animal experiments [30]. Of the NF-κB-involved cascades, the roles of iNOS and COX-2 downstream of NF-κB phosphorylation were well established by several previous studies [31]. iNOS uses NADPH and oxygen molecules to synthesize NO from L-arginine, resulting in oxidative stress and inflammatory response [32]. According to Comer et al. [33], acceleration of asthma onset is caused by abnormal expression of type 2 cells, which is caused by excessively produced NO, and iNOS is a major factor in synthesizing it [34]. In asthmatic patients, iNOS expression increased concurrently with the upregulation of NF-κB expression [35]. COX-2 is also thought to be a protein that plays a major role in the inflammatory response, a major symptom of asthma. Increased COX-2 levels induce prostaglandin production, leading to an inflammatory response [36]. The protective effects of natural products in asthmatic models through the regulation of expression of iNOS and COX-2 are well established [37,38]. In this study, phosphorylation of NF-κB and expression levels of iNOS and COX-2 proteins were significantly increased in the OVA group. In contrast, consistent with a previous study that showed the inhibitory effects of CIE against NF-κB phosphorylation, administration of CIE significantly reduced NF-kB, iNOS, and COX-2 compared with the OVA group in a dose-dependent manner [13]. Combined, these results suggest that CIE effectively suppresses the NF-κB pathway, leading to airway inflammation in allergic asthma.

MMP-9 is a proteolytic enzyme whose main function is to break down the extracellular structure and proteolytic modulation, resulting in tissue remodeling [39]. The inflammatory response following the production of cytokines and growth factors plays a major role in the expression of MMP-9 [40]. Various studies have demonstrated that MMP-9 plays a major role in the onset and exacerbation of asthma [41]. Cataldo et al. [42] found, through clinical tests, that MMP-9 levels were significantly higher than normal in the BALF, blood and sputum of asthma patients. McMillan et al. [43] experimentally demonstrated that airway inflammation and mucus secretion were reduced in MMP-9-deficient mice compared to normal mice in an asthma-induced experiment. MMP-9 is modulated by iNOS expression, which results in an asthmatic response via the upregulation of type 2 cytokine production [18]. When the expression level of iNOS, which is closely related to the inflammatory response, increases, MMP-9 sequentially increases, which results in degradation of the extracellular structure of the lung tissue and remodeling of the airways [44]. In addition, Huang et al. [45] demonstrated that increased COX-2 expression in a carbon tetrachloride-induced liver injury model results in increases in MMP-9 expression. In this study, CIE groups showed markedly reduced levels of MMP-9 in lung tissues compared with those of the OVA group revealed by IHC, Western blotting, and zymography. These results indicate that CIE in turn effectively inhibits MMP-9 expression, thereby reducing the type 2 cytokine effect and potentiating inflammatory response in allergic asthma.

## 5. Conclusions

In conclusion, CIE effectively reduced the inflammatory response induced by OVA sensitization/challenge. This effect was closely related to the suppression of NF-κB, iNOS, COX-2, and MMP-9 expression, followed by the production of type 2 cytokines and OVA-specific IgE. It relieved asthma symptoms, including AHR, airway inflammation, and mucus overproduction. Our results suggest that CIE is a potent therapeutic agent for the treatment of asthma.

## Figures and Tables

**Figure 1 nutrients-14-02067-f001:**
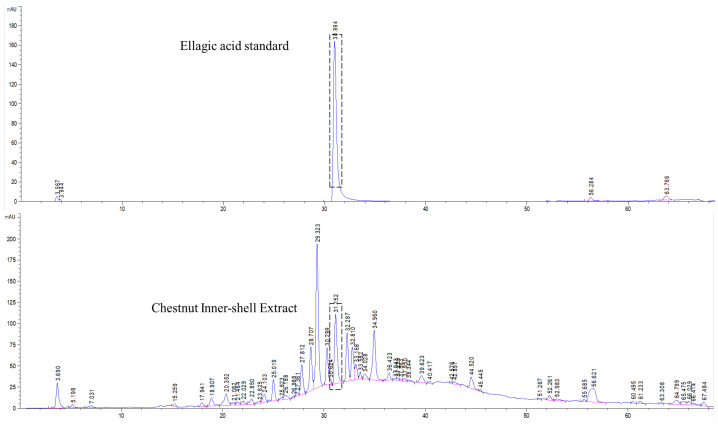
HPLC analysis of chestnut inner shell extract (CIE) was carried out with a UV detector for quantitative analysis of the major components.

**Figure 2 nutrients-14-02067-f002:**
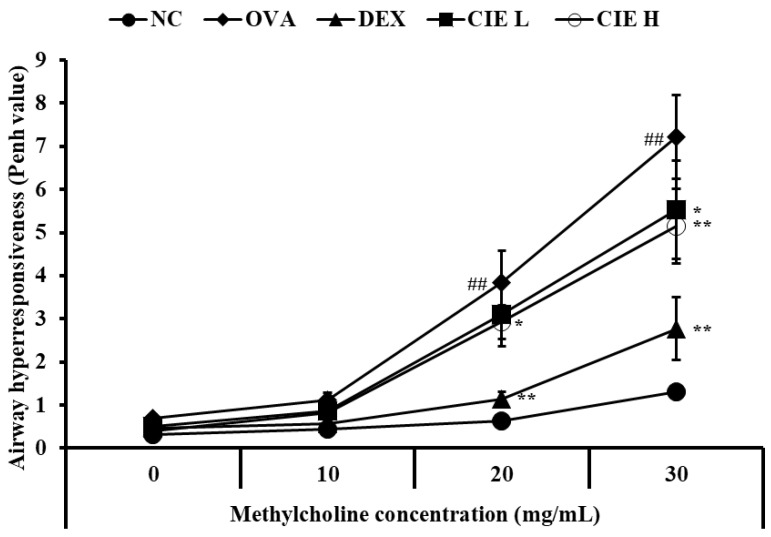
Chestnut inner shell extract (CIE) decreased the elevated airway hyperresponsiveness (AHR). Normal control (NC): mice treated with phosphate-buffered saline only; ovalbumin (OVA): mice sensitized and challenged with OVA; dexamethasone (DEX): mice sensitized and challenged with OVA and treated with dexamethasone (10 mg/kg); CIE L and H: mice sensitized and challenged with OVA and treated with CIE (100 and 300 mg/kg, respectively). Values are presented as means ± SD (*n* = 6). Significantly different from NC, ## *p* < 0.01; significantly different from OVA, *, ** *p* < 0.05 and 0.01, respectively.

**Figure 3 nutrients-14-02067-f003:**
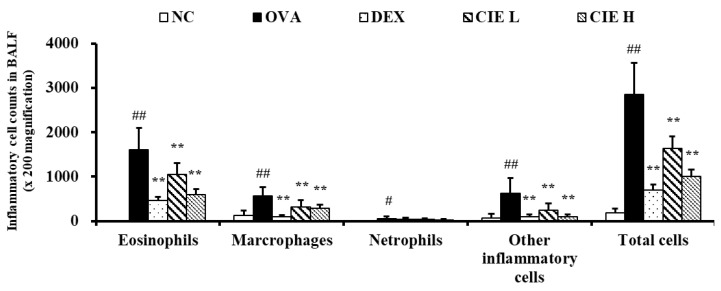
Chestnut inner shell extract (CIE) reduced the inflammatory cell count in bronchoalveolar lavage fluid. Normal control (NC): mice treated with phosphate-buffered saline only; ovalbumin (OVA): mice sensitized and challenged with OVA; dexamethasone (DEX): mice sensitized and challenged with OVA and treated with dexamethasone (10 mg/kg); CIE L and H: mice sensitized and challenged with OVA and treated with CIE (100 and 300 mg/kg, respectively). Values are presented as means ± SD (*n* = 6). Significantly different from NC, #, ## *p* < 0.05 and *p* < 0.01; significantly different from OVA, ** *p* < 0.01.

**Figure 4 nutrients-14-02067-f004:**
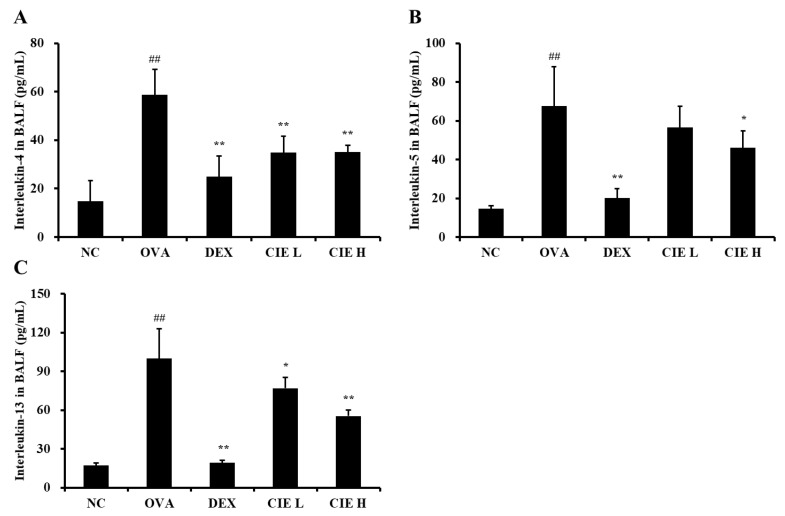
Chestnut inner shell extract (CIE) reduced the levels of (**A**) interleukin (IL)-4, (**B**) IL-5, and (**C**) IL-13 in the bronchoalveolar lavage fluid. Normal control (NC): mice treated with phosphate-buffered saline only; ovalbumin (OVA): mice sensitized and challenged with OVA; dexamethasone (DEX): mice sensitized and challenged with OVA and treated with dexamethasone (10 mg/kg); CIE L and H: mice sensitized and challenged with OVA and treated with CIE (100 and 300 mg/kg, respectively). Values are presented as means ± SD (*n* = 6). Significantly different from NC, ## *p* < 0.01; significantly different from OVA, *, ** *p* < 0.05 and 0.01, respectively.

**Figure 5 nutrients-14-02067-f005:**
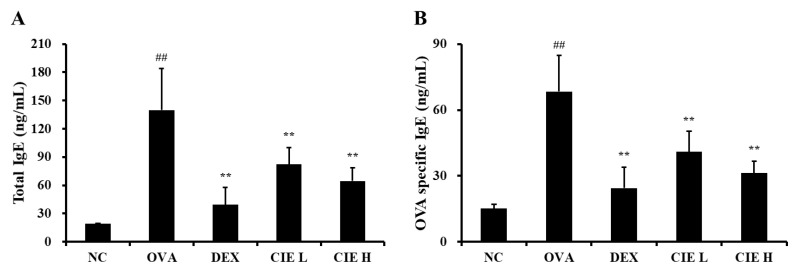
Chestnut inner shell extract (CIE) reduced the elevated levels of (**A**) total immunoglobulin E (IgE) and (**B**) ovalbumin (OVA)-specific IgE in the serum. Normal control (NC): mice treated with phosphate-buffered saline only; ovalbumin (OVA): mice sensitized and challenged with OVA; dexamethasone (DEX): mice sensitized and challenged with OVA and treated with dexamethasone (10 mg/kg); CIE L and H: mice sensitized and challenged with OVA and treated with CIE (100 and 300 mg/kg, respectively). Values are presented as means ± SD (*n* = 6). Significantly different from NC, ## *p* < 0.01; significantly different from OVA, ** *p* < 0.01, respectively.

**Figure 6 nutrients-14-02067-f006:**
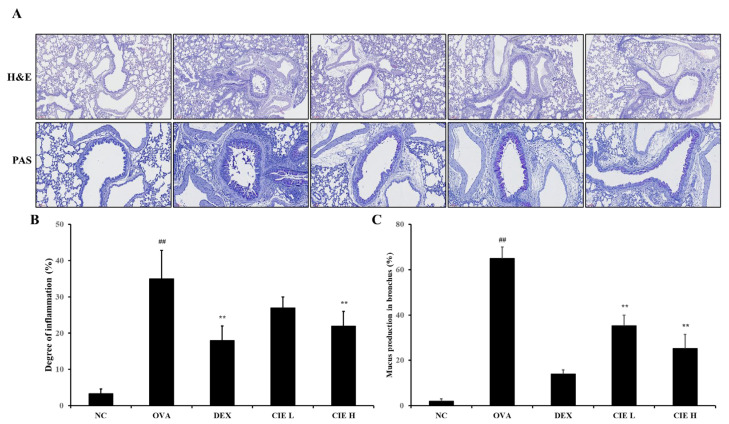
Chestnut inner shell extract (CIE) reduced the inflammatory responses and mucus production (**A**) in the lung tissue induced by OVA treatment. Quantitative analyses of airway inflammation (**B**) and mucus production (**C**) were performed utilizing an image analyzer. Normal control (NC): mice treated with phosphate-buffered saline only; ovalbumin (OVA): mice sensitized and challenged with OVA; dexamethasone (DEX): mice sensitized and challenged with OVA and treated with dexamethasone (10 mg/kg); CIE L and H: mice sensitized and challenged with OVA and treated with CIE (100 and 300 mg/kg, respectively). Values are presented as means ± SD (*n* = 6). Significantly different from NC, ## *p* < 0.01; significantly different from OVA, ** *p* < 0.01, respectively.

**Figure 7 nutrients-14-02067-f007:**
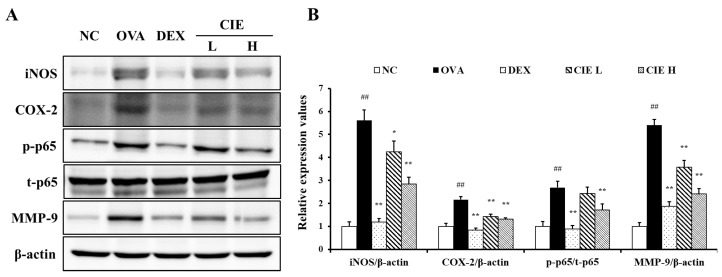
Chestnut inner shell extract (CIE) reduced the (**A**) expression of inducible nitric oxide synthase (iNOS), cyclooxygenase-2 (COX-2), matrix metalloproteinase-9 (MMP-9) and phosphorylation of nuclear factor kappa (NF-κB) in the lung tissues, of which (**B**) densitometric values were determined using Chemi-Doc. Normal control (NC): mice treated with phosphate-buffered saline only; ovalbumin (OVA): mice sensitized and challenged with OVA; dexamethasone (DEX): mice sensitized and challenged with OVA and treated with dexamethasone (10 mg/kg); CIE L and H: mice sensitized and challenged with OVA and treated with CIE (100 and 300 mg/kg, respectively). Values are presented as means ± SD (*n* = 6). Significantly different from NC, ## *p* < 0.01; significantly different from OVA, *, ** *p* < 0.05 and 0.01, respectively.

**Figure 8 nutrients-14-02067-f008:**
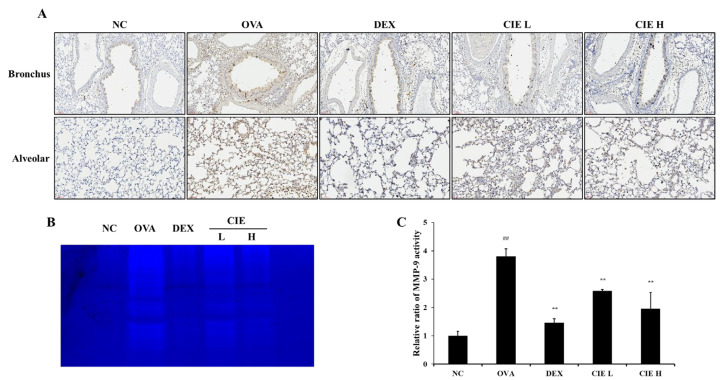
Chestnut inner shell extract (CIE) reduced the (**A**) expression of MMP-9 in lung tissue as determined by immunohistochemistry and (**B**) MMP-9 activity as determined by zymography. (**C**) Relative ratio of MMP-9 activity was determined using an image analyzer. Normal control (NC): mice treated with phosphate-buffered saline only; ovalbumin (OVA): mice sensitized and challenged with OVA; dexamethasone (DEX): mice sensitized and challenged with OVA and treated with dexamethasone (10 mg/kg); CIE L and H: mice sensitized and challenged with OVA and treated with CIE (100 and 300 mg/kg, respectively). Values are presented as means ± SD (*n* = 6). Significantly different from NC, ## *p* < 0.01; significantly different from OVA, ** *p* < 0.01, respectively.

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
