# Peer review of "Inner Shell of the Chestnut (Castanea crenatta) Suppresses Inflammatory Responses in Ovalbumin-Induced Allergic Asthma Mouse Model"

_nutrients, 2022, doi:10.3390/nu14102067_

Round 1

Reviewer 1 Report

The nutrients-1726669 addresses the potential preventive effects of inner shell components in chestnuts against asthma-related symptoms. The authors used an animal model and it is a strength of the nutrients-1726669. However, the nutrients-1726669 also require a decent update for better understanding for potential readers.

  1. [L16] Please clarify the injection route.
  2. [L17] Please provide the full chemical name instead of aluminum.
  3. [L32] One precedent paragraph is necessary as a first paragraph to explain 1) the asthma suffering population, 2) the growing reason for asthma suffering population, 3) the current treatment method, and 4) limitations of the current treatments (i.e. side effects).
  4. [L74] Detailed information is required for the HPLC analysis

- supplier information is required for C18 column

- autosampler temperature

- data acquisition methods

  1. [L137] Some primary antibodies do not have dilution factors and there is no information for secondary antibodies and chemiluminescence.
  2. [L162] statistically
  3. [Figure 1] Is this the novel finding?
  4. [L171] small letter for Chestnut (Coherence required through the entire manuscript).
  5. [L203] p-value in the result does not make sense. (i.e. p<0.1, 0.5).
  6. [Figure 4/6] y-axis is too small.
  7. [L265] t-65-actin?
  8. In discussion, authors should acknowledge by which functional components in the CIE with ADME (absorption, distribution, metabolism, and excretion) information.

Author Response

Reviewer #1: The nutrients-1726669 addresses the potential preventive effects of inner shell components in chestnuts against asthma-related symptoms. The authors used an animal model and it is a strength of the nutrients-1726669. However, the nutrients-1726669 also require a decent update for better understanding for potential readers.

  1. [L16] Please clarify the injection route.

- Thanks for your kind comments. We inserted the injection route (intraperitoneally) in abstract section as you pointed out.

  1. [L17] Please provide the full chemical name instead of aluminum .

- Thanks for your kind comments. We changed term ‘aluminum’ to ‘aluminum hydroxide’ as you pointed out.

  1. [L32] One precedent paragraph is necessary as a first paragraph to explain 1) the asthma suffering population, 2) the growing reason for asthma suffering population, 3) the current treatment method, and 4) limitations of the current treatments (i.e. side effects).

- Thanks for your kind comments. We inserted the paragraph which explaining what you pointed out. (the asthma suffering population, the growing reason for asthma suffering population, the current treatment method, and limitations of the current treatments).

  1. [L74] Detailed information is required for the HPLC analysis

- Thanks for your indication, we amended phrase by adding method details.

supplier information is required for C18 column

- Thanks for your kind comments. We checked supplier information for HPLC device and column and correct supplier information by adding ‘Agilent Technologies, Inc., Santa Clara, CA, USA’.

autosampler temperature

- Thanks for your kind comments. The column temperature was set at 40°C. Unfortunately, our HPLC system (Agilent 1260 infinity II) does not have thermos-control system in the autosampler.

data acquisition methods

- Thanks for your kind comments. Data acquisition was done by using the Agilent OpenLab CDS ChemStation (version 2.15.26).

  1. [L137] Some primary antibodies do not have dilution factors and there is no information for secondary antibodies and chemiluminescence.

- Thanks for your kind comments. We inserted the missed dilution factors of the primary antibodies. In addition, information of secondary antibody and chemiluminescence were added, and the company and address of the materials used in present experiment were thoroughly re-checked.

  1. [L162] statistically

- Thanks for your kind comments. We deleted the misspelled -.

  1. [Figure 1] Is this the novel finding?

- Thanks for your kind comments. Several active ingredients in a chestnut inner shell were previously revealed. However, the composition and the dominant contents in the compounds can be differed by the extraction method. In this study, we prepared standardized chestnut inner shell extract by using additional open column method to exclude highly water soluble fraction and increase ethanol soluble contents.

  1. [L171] small letter for Chestnut (Coherence required through the entire manuscript).

- Thanks for your kind comments. We corrected the uppercase letters you pointed out to lowercase letters and checked the entire paper for the same mistakes.

  1. [L203] p-value in the result does not make sense. (i.e. p<0.1, 0.5).

- Thanks for your kind comments. We changed incorrectly written 0,1 to 0.01 and 0.5 to 0.05.

  1. [Figure 4/6] y-axis is too small.

- Thanks for your kind comments. We resized figures 4 and 6 for readability increase.

  1. [L265] t-65-actin?

- Thanks for your kind comments. We deleted the misspelled -actin and inserted the modified figure.

  1. In discussion, authors should acknowledge by which functional components in the CIE with ADME (absorption, distribution, metabolism, and excretion) information.

- Thanks for your kind comments. As per reviewer’s indication, we amended manuscript by adding component and its ADME information at the discussion part.

Reviewer 2 Report

With a real pleasure, I read the manuscript nutrients-1726669.

My comments/remarks are only minor:

Comment 1. You refer to Th2 cytokines. It is correct but these days it is more common to call them type 2 cytokies (e.g. PMID: 33255348, 34836408).

Comment 2. Lines 91 to 101. It would be useful to add a scheme/timeline of the animal experiment procedure. In addition, sorry if I overlooked something, but when was CIE (and DEX?) delivered. Again, sorry if I overlloked something I can see it only for CIE only in the Abstract but not in the main text at the moment. It should be written (DEX as well) in the main text as well, methodological section.

Comment 3. Figure 2. OVA and CIE L symblos are too similar and thus misleading. Please, correct.

Comment 4. Figure 4. In this and some other figures are certain details too small to be seen well. Please, amend.

Comment 5. You show some downstream pathways tob e involved. Still, there is some more space for mechanistic considerations, at least discussions. Especially, polyphenol-induced epigenetic changes (PMID: 30823645, 33668787) could contribute to the effects observed by you. Please, discuss/speculate.

Comment 6. Graphical abstract would be welcome.

Author Response

Reviewer #2: With a real pleasure, I read the manuscript nutrients-1726669.

My comments/remarks are only minor:

Comment 1. You refer to Th2 cytokines. It is correct but these days it is more common to call them type 2 cytokies (e.g. PMID: 33255348, 34836408).

- Thanks for your kind comments. As you pointed out, we changed all the term ‘th2 cyokines’ to ‘type 2 cytokines’.

Comment 2. Lines 91 to 101. It would be useful to add a scheme/timeline of the animal experiment procedure. In addition, sorry if I overlooked something, but when was CIE (and DEX?) delivered. Again, sorry if I overlloked something I can see it only for CIE only in the Abstract but not in the main text at the moment. It should be written (DEX as well) in the main text as well, methodological section.

- Thanks for your kind comments. As you pointed out, we added the scheme/timeline of the animal experiment procedure in graphical abstract (Figure 9), and the duration and dosage of CIE and dexamethasone were additionally described in materials and methods section.

Comment 3. Figure 2. OVA and CIE L symblos are too similar and thus misleading. Please, correct.

- Thanks for your kind comments. We changed the similar symbols and resized them for better understanding.

Comment 4. Figure 4. In this and some other figures are certain details too small to be seen well. Please, amend.

- Thanks for your kind comments. We resized figures 4 and 6 for readability increase.

Comment 5. You show some downstream pathways tob e involved. Still, there is some more space for mechanistic considerations, at least discussions. Especially, polyphenol-induced epigenetic changes (PMID: 30823645, 33668787) could contribute to the effects observed by you. Please, discuss/speculate.

- Thanks for your kind comments. We added a few references to support the logically weak mechanism part. Also, we inserted the additional discussion about functional components which contribute the protective effects observed present study referring to the papers you pointed out.

Comment 6. Graphical abstract would be welcome.

- Thanks for your kind comments. As you pointed out, we added the graphical abstract in manuscript (Figure 9).
